



# Estimation and evaluation of hourly MetOp satellites GPS DCBs with two different methods

Linlin Li[1,2], Shuanggen Jin[1,2]

[1]Shanghai Astronomical Observatory, Chinese Academy of Sciences, Shanghai, 200030, China
[2]School of Astronomy and Space Science, University of Chinese Academy of Sciences, Beijing, 100049, China

*Correspondence to*: Shuanggen Jin (sgjin@shao.ac.cn)

**Abstract.** Differential code bias (DCB) is one of Global Positioning System (GPS) errors, which affects the calculation of total electron content (TEC) and ionospheric modeling. In the past, DCB was normally estimated as a constant in one day, while DCB of low Earth orbit (LEO) satellite GPS receiver may have large variations within one day due to complex space
environments and highly dynamic orbit conditions. In this study, daily and hourly DCBs of Meteorological Operational (MetOp) satellites GPS receivers are calculated and evaluated using spherical harmonic function (SHF) and local spherical symmetry (LSS) assumption. The results demonstrated that both approaches could obtain accurate and consistent DCB values. The estimated daily DCB standard deviation (STD) is within 0.1 ns in accordance with the LSS assumption, and is numerically less than the standard deviation of the reference value provided by COSMIC Data Analysis and Archive Center
(CDAAC). The average error's absolute value is within 0.2 ns with the respect to the provided DCB reference value. As for the SHF method, the DCB's standard deviation is within 0.1ns, which is also less than the standard deviation of the CDAAC reference value. The average error of the absolute value is within 0.2 ns. The estimated hourly DCB with LSS assumptions suggested that calculated results of MetOpA, MetOpB, and MetOpC are, respectively, 0.5 ns to 3.1 ns, -1.1 ns to 1.5 ns, and -1.3 ns to 0.7 ns. The root mean square error (RMSE) is less than 1.2 ns, and the STD is under 0.6ns. According to the SHF
method, the results of MetOpA, MetOpB, and MetOpC are 1 ns to 2.7 ns, - 1 ns to 1 ns, and - 1.3 ns to 0.6 ns, respectively. The RMSE is under 1.3 ns and STD is less than 0.5 ns. The STD for solar active days is less than 0.43 ns, 0.49 ns, and 0.44 ns, respectively, with the LSS assumption, and the appropriate fluctuation ranges are 2.0 ns, 2.2 ns, and 2.2 ns. The variation ranges for the SHF method are 1.5 ns, 1.2 ns, and 1.2 ns, respectively, while the STD is under 0.28 ns, 0.35 ns, and 0.29 ns.

## 1 Introduction

The ionosphere is an important part of the space environment, and the ionosphere observations and modeling are hot topics in space weather research. Although there have been quite a lot of studies on the ionosphere, the topside ionosphere is quite difficult to model due to the lack of directly observed data (Jin et al., 2021). At present, many low Earth orbit (LEO) satellites carried Global Positioning System (GPS) receivers for accurate orbit determination, and topside ionospheric total electron content (TEC) can be obtained by using the dual-frequency GPS data. However, accurate TEC estimation from LEO





satellite GPS observations is complicated due to the large number of effects or errors. The differential code bias (DCB) is

one of the errors in calculating TEC due to complex space environments and highly dynamic orbit conditions. The accuracy

can be as large as 20 TECU (7 ns) for each satellite and 40 TECU (14 ns) for the receivers when evaluating the TEC without

taking into account the DCB from the satellites and receivers (Abid et al., 2016). The GPS DCB and receiver DCB can be

estimated from the dual-frequency GPS observations (Sardon and Zarraoa, 1997; Arikan et al., 2008; Su et al., 2021).

Although DCB can be considered as an instrument hardware delay, the complex spatial environment prevents instrument

measurement in practice. As a result, some fast and reliable approaches to estimate the DCB of LEO GPS receivers are

required.

The DCBs are mainly estimated as an unknown parameter based on the non-geometric combination method and uncombined

precise point positioning (PPP) method (Jin et al., 2012; Zhang et al., 2011). The traditional geometry-free combination

approach (Zhang et al., 2011; Zhang et al., 2018) uses pseudo-range geometry free observation, phase geometry free

observation, and phase smoothing geometry free observation. Phase smoothing enhances the accuracy of pseudo-range non-

geometric measurement and avoids the estimate of ambiguity parameters in phase non-geometric measurement. According

to Jin et al. (2015) spherical harmonics can be used to simultaneously estimate the DCB of a ground-based receiver and a

GPS satellite. With an average difference of less than 0.7 ns and a root mean square error (RMSE) of less than 0.4 ns, the

results showed that the DCB computed by this technique has good consistency with the International Global Navigation

Satellite System (GNSS) Service (IGS) Analysis Center products. The uncombined PPP is the second approach to estimating

DCB (Zhang et al., 2011; Liu et al., 2018). By adding external limitations such as precise ephemeris and satellite clock offset,

the precision and dependability of non-different and uncombined PPP observation are increased (Zhang et al., 2011). The

precision is consistent with the phase smoothing pseudo-range method theoretically. Geometric-free observation will be

impacted by observation noise and the multi-path of pseudo-range code, but it can avoid the frequency-independent term and

dependency of outside constraint data. The computation procedure is rather straightforward, and observation accuracy will

steadily improve with an increase in smooth radian length. As a result, most GNSS ionospheric extraction mainly uses the

phase smoothing pseudo range geometry free method.

Yue et al. (2011) used the DCB of the GPS satellite supplied by IGS as the real value for the DCB of the LEO satellite

receiver and estimated the DCB of the LEO satellite receiver as the unknown parameter based on the spherical symmetry

assumption. Zhang et al. (2014) used the spherical harmonic function to parameterize the ionospheric TEC, and the least

square (LS) method to simultaneously estimate both ionospheric spherical harmonic function coefficients and DCB

parameters. The results revealed that the estimated values were in good agreement with the reference values. The root mean

square error (RMSE) value of the DCB difference was within 2 TECU, and the maximum absolute difference was less than 3

TECU. Lin et al. (2014) estimated the GPS satellite DCB and LEO satellite GPS receiver DCB simultaneously through

Constellation Observing System for Meteorology Ionosphere and Climate (COSMIC) and Challenging Minisatellite Payload

(CHAMP) data, and found that the median values of all satellites RMSE accuracy and mean precision from COSMIC and

CHAMP observations are 1.581 TECU and 0.235 TECU, 0.558 and 0.218 TECU, respectively. Wautelet et al. (2017)





estimated the DCB of JASON-2 with the local spherical symmetry assumption and showed that the solution of GPS satellite

DCB was very close to the solution of the IGS analysis center using ground measurement. Lin et al. (2023) used the dual
frequency observation data of three satellite GPS receivers in the SWARM constellation to estimate the DCB of GPS
satellites and LEO satellite GPS receivers. Compared with the independent estimation scheme, the stability of the GPS
satellite DCB obtained by the joint estimation scheme was 16.6% higher than that of the independent estimation scheme,
which had better consistency with the reference DCB. The GPS receiver DCB is calculated with the value utilized for

estimation by the product of the current receiver DCB and the vertical total electron content (VTEC) obtained from the
Global Ionosphere Map. However, the estimation of DCB is affected by TEC values, which may result in some
discrepancies between the estimated DCB and the true value, despite its higher precision.

DCBs for the satellite and receiver are commonly assumed as constants during a period of one day. That is, DCB can be
calculated using ionospheric features if DCBs have been found to be sufficiently stable in one day. Every single day, one

solution will be provided. However, DCBs cannot be assumed to be the same in one day if they experience some short-term
changes. Additionally, studies of receiver DCB fluctuation features over short time intervals should be estimated (Zhang et
al., 2015; Li et al., 2017; Xue et al., 2015). And although the DCBs of GNSS satellites are relatively stable, the DCBs of
LEO satellite GPS receivers may have obvious fluctuations due to various factors in highly dynamic orbits. The LEO
satellite GPS receiver DCB is more susceptible to the effects of the space environment and other factors than high-altitude

GNSS satellite DCB, and its stability is worse than that of high-altitude GNSS satellites at the same time. Thus, it is
necessary to estimate and analyze the LEO satellite GPS receiver DCBs in a short period.

In previous studies, many factors affected the stability of ground-based GNSS receiver DCB, such as the quality of the orbit
and observation data, space weather, the receiver type, antenna type, receiver hardware version, and receiver environment,
especially the temperature (Zhang et al., 2015; Xue et al., 2016; Xue et al., 2015; Li et al., 2016; Choi and Lee, 2018; Zha et

al., 2019). Analyzing the receiver DCB of the BeiDou navigation satellite system (BDS) and Galileo satellite navigation
system, it was found that the receiver type has no obvious relationship with the stability of the receiver DCB (Xue et al.,
2016; Xue et al., 2015; Li et al., 2016). But in the stability analysis of GPS receiver DCB, Choi and Lee (2018) found the
type of receiver and antenna have a certain influence. Meanwhile, they also found that after the receiver hardware version is
replaced, the receiver DCB value will change significantly for the receiver DCB of GPS, BDS, Galileo, and other systems

(Choi and Lee, 2018). There exists a strong linear correlation between the estimated receiver DCB and measured temperature
values (Zha et al., 2019). On ground-based receivers, the standard deviation (STD) of some receiver DCBs can reach 1-2 ns
(Wang et al., 2020). In space environments, when LEO satellites are moving, the temperature can change greatly, which may
cause great instability of the LEO satellite GPS receiver DCB. An error of up to 8 TECU may affect the computed DCB
during periods of strong solar activity. The estimated DCB may have an accuracy of about 3 TECU for low solar activity. In

comparison to the receiver DCB, the satellite DCB is more than ten times smaller (Conte et al., 2011). Furthermore, Kao et
al. (2013) claimed that estimating errors rather than DCB changes are to blame for some of the bigger daily deviations in



receiver DCBs. Various data processing techniques will result in various estimation mistakes. For some locations, smoothed and unsmoothed observations show DCB discrepancies of up to 6–8 TECU.

The Meteorological Operational (MetOp) satellites are in sun-synchronous near-circular orbit, and the ascending altitude is

from 796 km to 844 km (Maybeck, 1982). The MetOp mission consists of three satellites in orbit, and the height of the satellites is about 817 km. The MetOp mission is considered the first step for the Earth observation space segment of the Global Monitoring for Environment and Security (GMES) initiative. The COSMIC Data Analysis and Archive Center (CDAAC) offers orbital data, approximated LEO satellite receiver DCB data, and dual frequency GPS observation data onboard LEO satellites. Based on the local spherically symmetric assumption, the CDAAC uses geometric mapping

functions and the local spherical symmetry to determine the receiver DCB (Yue et al., 2011), whose accuracy is about 1-2 TECU. The three MetOp satellites data are available from 2019195 to 2021331. However, the MetOp satellite receiver DCB is rarely studied, particularly in a short period.

In this paper, daily and hourly DCBs of MetOp satellite receivers are estimated using spherical harmonics function (SHF) and local spherical symmetry (LSS) assumption, which are further evaluated and compared with the DCB products provided

by CDAAC. The MetOp data, local spherical symmetry assumption, and spherical harmonic function are introduced in Section 2. The results and analysis are presented in Section 3. In Section 4, the discussions are presented, and the conclusion is given in Section 5.

## 2 Data and Methods

This part introduces the data used and provides details on the LSS assumption and SHF method.

### 2.1 LEO Data

LEO satellites are easily affected by space weather. In order to reduce this effect and calculate the DCBs, a chosen period must satisfy the following conditions:

1. LEO observation data, the LEO satellites, and GPS satellite orbit data are available.

2. The observation period is as long as possible.

The time periods from September 9 (day of the year 252) 2019 to September 18 (day of the year 261) and September 5 (day of the year 248) to September 11 (day of the year 254) in the years 2021 and 2022, respectively, have been chosen. Figure 1 shows the solar activity and geomagnetic index during this time period. The F10.7 is under 80, the Dst is above -30 nT and Kp is under 4, which indicated in this period, the geomagnetic condition is calm and the solar activity is not quite active. The same concept also applies to Figure 2 during the solar active period, as was previously mentioned. The range of the F10.7 is

92–104, which suggests that this is a solar active period. (https://www.sws.bom.gov.au/Educational/1/2/4). Figure 3 illustrates the orbital paths of the three LEO satellites and their period of motion is 1.68 hours.



## 2.2 STEC estimation from GNSS

The Receiver Independent Exchange (RINEX) format is used to record carrier phase and pseudo-range measurements for GNSS. The pseudo-range and carrier phase observation equations for GPS are shown below (Jin et al., 2012):

$$P_{k,j}^i = \rho_{0,j}^i + d_{ion,k,j}^i + d_{trop,k,j}^i + c(\tau^i - \tau_j) + d_k^i + d_{k,j} + \varepsilon_{P,k,j}^i , \tag{1}$$

$$L_{k,j}^i = \rho_{0,j}^i - d_{ion,k,j}^i + d_{trop,k,j}^i + c(\tau^i - \tau_j) - \lambda(b_{k,j}^i + N_{k,j}^i) + \varepsilon_{L,k,j}^i , \tag{2}$$

where P and L are the GPS pseudo-range and phase measurement, respectively, $\rho$ is the distance between the GPS satellite and GPS receiver, $d_{ion}$ and $d_{trop}$ are ionospheric and troposphere delay, severally, c is the speed of light in vacuum environment, $\tau^i$ and $\tau_j$ are the satellite and receiver clock error, separately, $d$ is the code delays for the satellite and receiver

biases, N is the ambiguity of the carrier phase, and $\varepsilon$ is the other error in the GPS measurement. The phase advance of the satellite and receiver instrument biases can be represented by b.

The frequency is denoted by the subscript $k$ (=1, 2), the GPS receiver's sequence number is denoted by the subscript $j$, and the GPS satellite's sequence number is denoted by the superscript $i$. The ionospheric delays can be calculated from dual-frequency GPS measurements ($f_{L1} = 1575.42$MHz, $f_{L2} = 1227.60$MHz) with the following equations:

$$P_4 = P_{1,j}^i - P_{2,j}^i = d_{ion,1,j}^i - d_{ion,2,j}^i + DCB^i + DCB_j , \tag{3}$$

$$L_4 = L_{1,j}^i - L_{2,j}^i = -(d_{ion,1,j}^i - d_{ion,2,j}^i) - \lambda(N_{1,j}^i - N_{2,j}^i) , \tag{4}$$

where $DCB^i = d_1^i - d_2^i$ and $DCB_j = d_{1,j} - d_{2,j}$ stand for the differential code biases of the satellites and differential code biases of the receivers, respectively.

Due to the high noise in the pseudo-range observations $P_4$, the carrier phases are used to smooth the pseudo-range. The $P_{4,sm}$

is expressed after smoothing as follows:

$$P_{4,sm} = \omega_t P_4(t) + (1 - \omega_t)P_{4,prd}(t) \quad (t > 1) , \tag{5}$$

where t stands for the epoch number, $\omega_t$ is the weight factor related with epoch (Yuan et al., 2021). And

$$P_{4,prd} = P_{4,sm}(t - 1) + (L_4(t) - L_4(t - 1)) \quad (t > 1) , \tag{6}$$

The following function is an expression for the ionospheric delay:

$$d_{ion} = \frac{40.3}{f^2} STEC , \tag{7}$$

where $f$ stands the frequency of the carrier, and STEC stands for the slant total electron content.

With replacing $P_4$ by $P_{4,sm}$, we can get the following function:

$$P_{4,sm} = 40.3 \left(\frac{1}{f_1^2} - \frac{1}{f_2^2}\right) STEC + DCB^i + DCB_j , \tag{8}$$

Combining equations (7) and (8), the STEC from GNSS dual-frequency observations can be calculated as follow:

$$STEC = -\frac{f_1^2 f_2^2}{40.3(f_1^2 - f_2^2)} (P_{4,sm} - cDCB^i - cDCB_j) , \tag{9}$$

where $DCB$ unit is the time.



## 2.3 Mapping Function

The mapping function (MF) can convert STEC to VTEC. Compared with the single-layer MF used by ground-based observations (Zhong et al., 2015), the F&K geometric MF, whose geometric relation is shown in Figure 3, has a more reasonable performance of STEC and VTEC conversion for the GPS-LEO link (Foelsche and Kirchengast, 2002). The F&K geometric MF (Schaer et al., 1999) is expressed as:

$$\text{VTEC} = \text{MF(z)} * \text{STEC} ,\tag{10}$$

$$M_{F\&K}(z) = \frac{1+R}{\cos z + \sqrt{R^2 - (\sin z)^2}},\tag{11}$$

$$R = \frac{R_e + H_p}{R_e + H_l},\tag{12}$$

where $z$ refers to the zenith angle, $R_e$ is the radius of the Earth, $H_l$ is the orbit altitude of LEO satellites, and $H_p$ is the altitude of the single layer of ionospheric pierce point (IPP).

## 2.4 Spherical Harmonic Function

The spherical harmonic function (SHF) is an easy way to establish global VTEC map. The VTEC can be described as (Liu et al., 2020):

$$E(\beta, s) = \sum_{n=0}^{n_{max}} \sum_{m=0}^{n} \tilde{P}_{nm}(\sin(\beta))(a_{nm} \cos(ms) + b_{nm} \sin(ms)) ,\tag{13}$$

where $\beta$ is the geocentric latitude of the IPP, $s$ is the longitude of the IPP, $a_{nm}$ and $b_{nm}$ are the worldwide or regional ionosphere model coefficients, $\tilde{P}_{nm}$ are normalized Legendre polynomials.

The following equation can be established using equation 12 and equation 13:

$$\sum_{n=0}^{n_{max}} \sum_{m=0}^{n} \tilde{P}_{nm}(\sin(\beta))(a_{nm} \cos(ms) + b_{nm} \sin(ms)) =$$

$$\cos(\arcsin(\frac{R}{R+H}\sin(\alpha z)))(-\frac{f_1^2 f_2^2}{40.3(f_1^2 - f_2^2)}(P_{4,sm} - cDCB^i - cDCB_j)) ,\tag{14}$$

The order of the spherical harmonic expansion depends on the area. Here, a set of ionospheric coefficients every 4 hours is set based on the amount of collected data. In this paper, the order is set to be 8.

## 2.5 Local Spherical Symmetry Assumption Method

If the ionosphere is assumed as locally spherical symmetry and then the local spherical symmetry (LSS) assumption equation for a given observation epoch can be written as follows with n GPS satellites observed simultaneously by the onboard GPS receiver:

$$\begin{pmatrix} P^1 - c * DCB^1 \\ P^2 - c * DCB^2 \\ \vdots \\ P^n - c * DCB^n \end{pmatrix}_{n*1} = \begin{pmatrix} MF^1 & 1 & 0 & 0 \\ MF^2 & 0 & 1 & 0 \\ \vdots & \vdots & \vdots & \vdots \\ MF^n & 0 & 0 & 1 \end{pmatrix}_{n*4} * \begin{pmatrix} VTEC \\ DCB_A \\ DCB_B \\ DCB_C \end{pmatrix}_{4*1} ,\tag{15}$$



where MF is the mapping function. For the accuracy of calculation, the weight is set related to the GPS satellites elevation. Using the SHF method or LSS assumption, LEO receiver DCB and ionospheric coefficients or VTEC can be estimated by

the least square method. In fact, it is also possible to estimate the GPS and receiver DCB simultaneously through certain constraints (Liu et al., 2020). Although GPS DCB and LEO satellite DCB can be estimated at the same time, because the number of MetOp satellites is 3, GPS DCB data from DLR is used to reduce the number of unknown parameters to ensure accurate estimation of LEO satellite DCB.

**2.6 Error estimation method**

The error is represented by STD and RMSE in this article. RMSE, which measures how much the measured data deviates from the true value, is the square root of the ratio of the variation between the observed value and the true value to the number of observations N. A lower value denotes higher precision. If a CDAAC reference is available, the function below can be used to calculate RMSE.

$$X_{RMSE} = \sqrt{\frac{\sum_{i=1}^{N}(X_{cal,i}-X_{reference,i})^2}{N}} \; , \tag{16}$$

where $N$ is the 24 for one day estimation data, $X_{cal}$ is the calculated value and $X_{reference}$ is the reference from CDAAC. The average square of the discrepancy between each sample value and the mean of all sample values is the variance. The value of X is more steady the lower the deviation is. The mathematical square root of the variance is called STD. It may also reflect a dataset's degree of dispersion. The STD is used in the absence of a reference value and can be calculated using Eq. 17.

$$X_{STD} = \sqrt{\frac{\sum_{i=1}^{N}(X_{cal,i}-X_{mean})^2}{N}} \; , \tag{17}$$

where $X_{mean}$ is the mean value in one day calculation.

**3 Results and Analysis**

**3.1 Daily DCB Estimation**

First, the DCB is assumed as constant in a day, and the daily DCB estimation values for different time periods are estimated

and shown in Figure 4 and Figure 5. The reference values are provided by CDAAC. Focus on solar quiet days, the results of MetOpA are underestimated. The results for MetOpB are occasionally overstated and occasionally underestimated. Most of the calculated results are overestimated for MetOpC. But the average error absolute value for the two methods is within 0.17 ns for LSS assumption and within 0.16 ns for SHF method, respectively. While the absolute value of the mean error of MetOpB is the smallest among the three satellites. Although there are some differences in numerical values, the trend shows

consistency. Unfortunately, there are no reference values for solar active days. Figure 6 provides the calculated values from LSS and SHF. And the average error absolute values between the two methods are within 0.04 ns and 0.24 ns for solar quiet



days and solar active days, respectively. Within the solar quiet days, the reference value of DCB is quite stable, and the STD is within 0.11 ns. As for the estimation result from the LSS assumption, the calculated DCB STD is within 0.08 ns with respect to the reference value. The results of the SHF method demonstrate that the computed DCB standard deviation is within 0.07 ns. And within the solar active days, the STD of LSS calculated DCB is within 0.10 ns and the STD of SHF calculated DCB is within 0.09 ns. Some error analysis results are provided in Table 1 in full detail.

Both approaches are excellent within the allowable error range, as can be seen from the study and comparison that was done above. Both independent methods can obtain consistent receiver DCB results with respect to the CDAAC. The results of the study also clearly support the reliable MetOp receiver DCB values offered by CDAAC. Besides, there are some differences between the estimated results of the two methods and the reference values, which may be due to the method error.

**3.2 Hourly DCB Estimation on Solar Quiet Days**

Hourly space-based GPS receivers DCBs are further estimated and shown in the figures below. Figure 7 illustrates the estimated hourly DCB from LSS assumption, and Figure 9 shows RMSE and STD of hourly DCB estimation from LSS assumption based on MetOpA, MetOpB, and MetOpC, respectively. For the estimation results based on LSS assumptions, it can be found that the calculation results of MetOpA, MetOpB, and MetOpC range from 0.5 ns to 3.1 ns, -1.1 ns to 1.5 ns, and -1.3 ns to 0.7 ns, respectively. The RMSE ranges from 0.8 ns to 1.2 ns, 0.7 ns to 1.1 ns, and 0.5 ns to 1 ns, respectively. And the STD ranges from 0.3 ns to 0.5 ns, 0.2 ns to 0.6 ns, and 0.2 ns to 0.5 ns, respectively.

The estimated DCBs from the SHF method are shown in Figure 8 as well as their RMSE and STD in Figure 10. The calculation results show that the hourly DCBs have a certain change, while almost the calculated DCB is less than the CDAAC-provided reference value. For the estimation results from the SHF method, the calculation results of MetOpA, MetOpB, and MetOpC range from 1 ns to 2.7 ns, - 1 ns to 1 ns, and - 1.3 ns to 0.6 ns, respectively. The RMSE ranges from 1.1 ns to 1.3 ns, 0.9 ns to 1.2 ns, and 0.7 ns to 1.2 ns, respectively. And the STD is from 0.2 ns to 0.4 ns, 0.2 ns to 0.4 ns, and 0.2 ns to 0.4 ns, respectively.

The average value of the error between DCBs from LSS and SHF and the given reference value in one day is shown in Figure 11. With the LSS assumptions, the average daily error ranges from -1.8 ns to 0.3 ns, -0.9 ns to -0.1 ns, and -0.9 ns to -0.1 ns, respectively. And with the SHF method, the average daily error ranges from - 1.5 ns to -0.7 ns, - 1.8 ns to -0.7 ns, and - 1.2 ns to -0.6 ns, respectively.

Compared to LSS results, SHF outputs are still more precise and stable. But the SHF method requires a larger amount of data because there are more unknown parameters, which is also the limitation of this method. The main difference between the two calculation methods in calculating the daily DCB is not very large. But when calculating the hourly DCB, the results of the two methods are quite different. Therefore, when the amount of data is not enough, the LSS assumption is much better to calculate DCB. If the amount of data is enough, the SHF method is more recommended. Hourly DCB time series shows high-frequency variations in one day. According to the statistical chart of frequency error analysis in Figure 12, the error conforms to a normal distribution. Each $\mu$ is under 0 ns, and the $\sigma$ is under 0.42 ns. This means that overall deviation is




existed, which may come from the method error. And the change of the calculated DCB should be attributed to the random error. In addition, the reference DCB from CDAAC is not precise. And the used MF can also cause an error between the calculated result and the reference value. Therefore, more errors and causes should be further studied and discussed in the future.

**3.3 Hourly DCB Estimation on Solar Active Days**

Hourly space-based receiver DCBs are further calculated. As shown in Figure 13, the daily fluctuation for MetOpA using the LSS approach is around 2 ns. Additionally, the highest fluctuations for MetOpB and MetOpC are approximately 2.2 ns. SHF is used, and Figure 14 displayed that for the three LEO satellites, the changes in DCB are, respectively, 1.5 ns, 1.2 ns, and 1.2 ns. Unfortunately, there is no reference value at this time. Thus, the STD is computed. The STD of DCB from LSS method is shown in Figure 15 for MetOpA, MetOpB, and MetOpC, respectively. The range for MetOpA is 0.28 to 0.43 ns.
For MetOpB, the range is 0.23 to 0.49 ns. As for MetOpC, the range is from 0.29 ns to 0.44 ns. Figure 16 shows the results of the SHF method. For MetOpA, the STD ranges from 0.21 ns to 0.28 ns. The range for MetOpB is between 0.2 and 0.35 ns. For MetOpC, the STD ranges from 0.21 to 0.29 ns.

**4 Discussion**

In this study, the LSS assumption and SHF method are used respectively to calculate the GPS receiver DCB of the LEO
satellite in different solar conditions. The STD of the daily DCB calculated according to the LSS assumption for 9-18 September, 2020 is within 0.08 ns and is numerically smaller than the standard deviation of the reference value provided by CDAAC. The absolute value of the average error is within 0.17 ns with respect to the DCB reference value. For solar active days, although there is no reference value from CDAAC, the STD of the two methods can be introduced. And the results show that the STD is within 0.10 ns for the LSS assumption and is within 0.09 ns for the SHF method. The results show that
LSS and SHF can be used to calculate daily DCBs.

    The two methods are applied to calculate the hourly DCBs. For solar quiet days, the RMSE of MetOpA, MetOpB, and MetOpC from LSS is below 1.2 ns, 1.1 ns, and 1 ns, respectively. The STD of MetOpA, MetOpB, and MetOpC are below 0.5 ns, 0.6 ns, and 0.5 ns, respectively. And the average daily error of MetOpA, MetOpB, and MetOpC ranges from -1.8 ns to 0.3 ns, -0.9 ns to -0.1 ns, and -0.9 ns to -0.1 ns, respectively, which are higher than the daily estimation values. As for the
RMSE of MetOpA, MetOpB, and MetOpC from the SHF method, they are less than 1.3 ns, 1.2 ns, and 1.2 ns, respectively. The MetOpA, MetOpB, and MetOpC STD are below 0.35 ns, 0.40 ns, and 0.36 ns, respectively. The average daily error of MetOpA, MetOpB, and MetOpC ranges from -1.5 ns to -0.7 ns, -1.8 ns to -0.7 ns, and -1.2 ns to -0.6 ns, respectively, which are also higher than the daily calculated DCBs. The fluctuations in one day of LSS and SHF are less than 2.24 ns and 1.62 ns, respectively. For solar active days, the STD of MetOpA, MetOpB, and MetOpC from LSS are, respectively, 0.37 ns, 0.39 ns,
and 0.36 ns. And from SHF, the STDs of MetOpA, MetOpB, and MetOpC are less than 0.26 ns, 0.27 ns and 0.26 ns. For the



three LEO satellites, the fluctuations in one day of LSS and SHF are less than 2.20 ns and 1.53 ns, respectively. In other words, the stability and change of DCBs on solar active days and solar quiet days are similar. Besides, the accuracy of hourly DCB estimation in solar active days cannot be debated because CDAAC did not provide references from 20210905-20210911.

Therefore, both methods can calculate reliable DCB results, whether DCB is assumed as the same in one day or only in one hour. And it is easy to find that compared to a day's data result, an hour's estimation error is much larger, which is the same as found in the previous studies (Li et al., 2017). The estimation accuracy of DCB is related to the amount of data as mentioned in introduction. This is also the reason why the calculation error of the daily estimations is smaller. Although the estimation results from the LSS assumption and the SHF method are slightly different, they are both stable and reliable,

while the SHF method is a little more precise.

The hourly DCB estimation results show a certain change in the LEO satellite GPS receiver DCB. Although the error looks large, according to Choi and Lee (2019), the ground-based GNSS receiver DCBs can reach tens of nanoseconds. The STD of receiver DCB at all stations is below 2 ns, and Wang et al. (Wang et al., 2020) also obtained the same results, implying that ground-based receiver DCB is less stable than satellite DCB. As previously indicated, the absolute error for the daily LEO

satellite receiver DCB estimation is 3 TECU (or around 1.05 ns) (Zhang and Tang, 2014). Our hourly DCB estimation results for receivers on LEO satellites are accurate. Comparing the DCBs under the two scenarios, it is hard to demonstrate that the space weather can affect the results of DCB estimation. The DCB of the GPS receiver sometimes has significant intraday changes, which may be due to fluctuations in ambient temperature as introduced before. But in this study, the calculated hourly DCBs do not change greatly whether in solar calm days or in solar active days. Although the MetOp

satellite can rotate around the Earth once every approximately 1.6 hours and the space temperature changes apparently, the change of receivers DCB has little with the environment temperature because the temperature control systems in the satellites exist. But the temperature inside the satellites related to the hourly change of ground-based DCB is unknown because there is a lack of detailed data on the temperature inside the satellites. As in the previous study, it was found that the receiver CPU temperature also affected the DCB (Yue et al., 2011). Therefore, it might also be a reason for the change of

LEO satellite GPS receiver DCB.

In addition, the antenna types, hardware, etc. may also cause DCB changes. But for MetOpA, MetOpB, and MetOpC, each LEO satellite uses same kind of antenna to receive signals. Therefore, it is excluded the antenna type influence during the calculation because the hardware conditions are unified. According to the statistical analysis of frequency errors in Figure 12, the error conforms to a normal distribution, and the error may be the main reason for the change in the calculated DCB.

In data processing, this paper only estimates the receiver DCBs of three MetOp satellites. In the future, more satellites with similar heights should be used to estimate DCB, which can increase the data volume and improve the overall estimation accuracy (Choi and Lee, 2019).

## 5 Conclusion

In this study, the LSS assumption and SHF method can both calculate the LEO satellite GPS receiver DCB well. It also
provided verification for reference values from CDAAC. Besides, the LEO satellite receiver DCBs provided by CDAAC are
missing on some dates, so we can also calculate the missing DCB value of the LEO satellite GPS receiver with these two
methods. In addition, we also calculated and analyzed the hourly DCB through these two methods, and the main conclusions
are summarized as follows:

1. The LSS assumption and SHF method can both estimate the reliable LEO GPS receiver DCB well, whether DCB is
assumed as the same or different in one day;

2. The SHF method is more stable and precise than the LSS assumption when compared with the reference value provided
by CDAAC. Also, the daily DCB estimation is more accurate and stable than the hourly DCB due to the more amount of
data;

3. Hourly DCBs have changes in one day, but these can mainly be attributed to random errors because these error time series
conform a normal distribution. Satellite internal temperature may also be a possible reason to cause the change of hourly
receiver DCB.

*Data availability*. The F10.7, Kp and Dst index can be obtained at https://spdf.gsfc.nasa.gov/. The MetOp satellites
observations data from CDAAC is available at https://cdaac-www.cosmic.ucar.edu/. The precise orbit products from CDDIS
are available at https://cddis.nasa.gov/Data_and_Derived_Products/GNSS/. The GPS DCB products provided by DLR can
be obtained at https://cddis.nasa.gov/archive/gnss/products/bias/.

*Author contributions*. Conceptualization of Manuscript Idea: S.J. and L.L. Methodology and Software: L.L; Supervision and
Funding Acquisition: S.J.; L.L. wrote the original draft preparation; S.J. reviewed and edited this paper. All authors have
read and agreed to the published version of the manuscript.

*Competing interests*. The contact author has declared that neither they or their co-authors have any competing interests.

*Acknowledgement*. The authors thank National Aeronautics and Space Administration (NASA)'s Space Physics Data Facility
for providing the space weather index, CDAAC for providing MetOp satellites products, IGS for providing GNSS precise
orbit products, and DLR for providing GPS DCB products.

*Financial support*: This research was funded by the National Natural Science Foundation of China (NSFC) Project, grant
number 12073012.






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

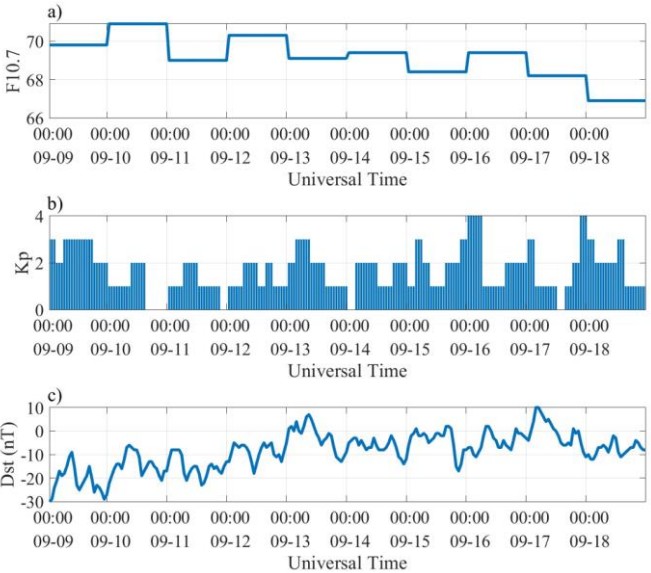

**Figure 1: Space weather condition with F10.7 (a), Kp (b) and Dst (c) from 20190909 to 20190918. The LEO satellite receiver data in three MetOp satellites are selected from CDAAC.**

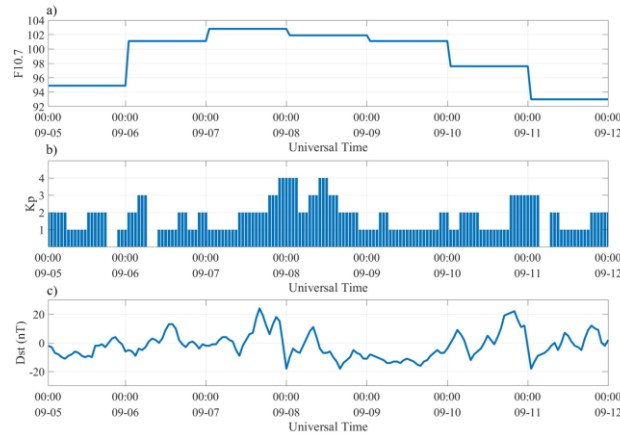


**Figure 2: Space weather condition with F10.7 (a), Kp (b) and Dst (c) from 20210905 to 20210911. Unfortunately, CDAAC does not have the LEO satellite receiver DCB value in this period, which is also a problem for MetOp satellite.**





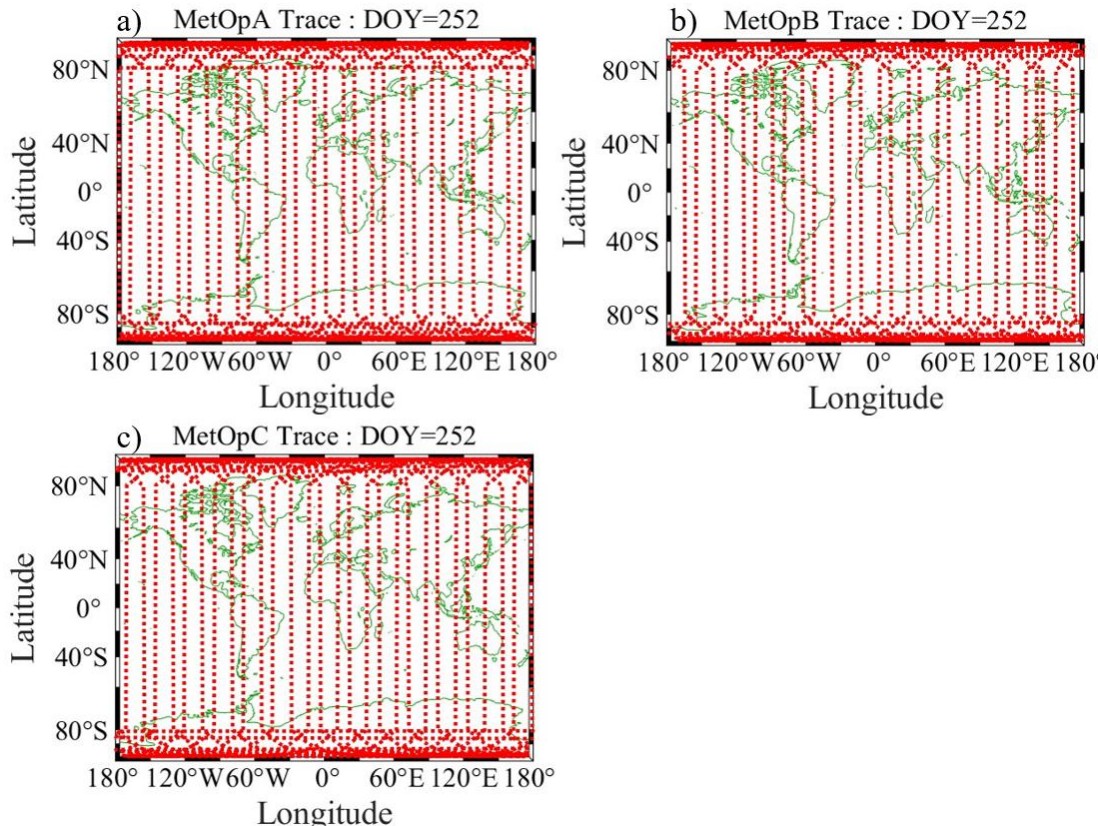

**Figure 3: Satellite traces of MetOpA (a), MetOpB (b) and MetOpC (c) on DOY 252, 2019.**

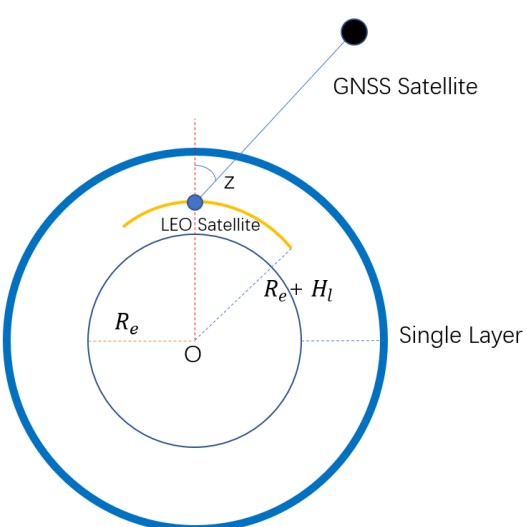


**Figure 4. F&K geometric MF.**





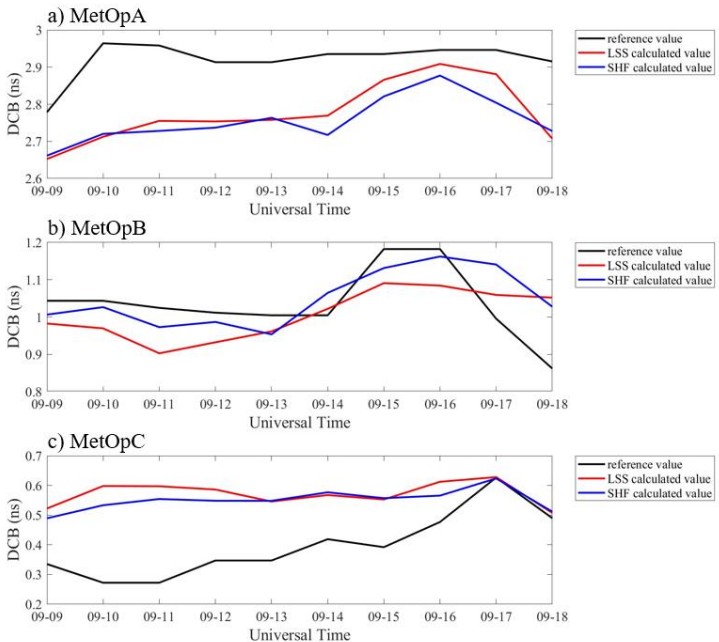

**Figure 5. DCB sequence of MetOp satellites GPS receivers from September 9 to September 18, 2019. The black line represents the receiver DCB provided by CDAAC, the red line represents the receiver DCB estimated through LSS, and the blue line is the**
**receiver DCB estimated through SHF.**

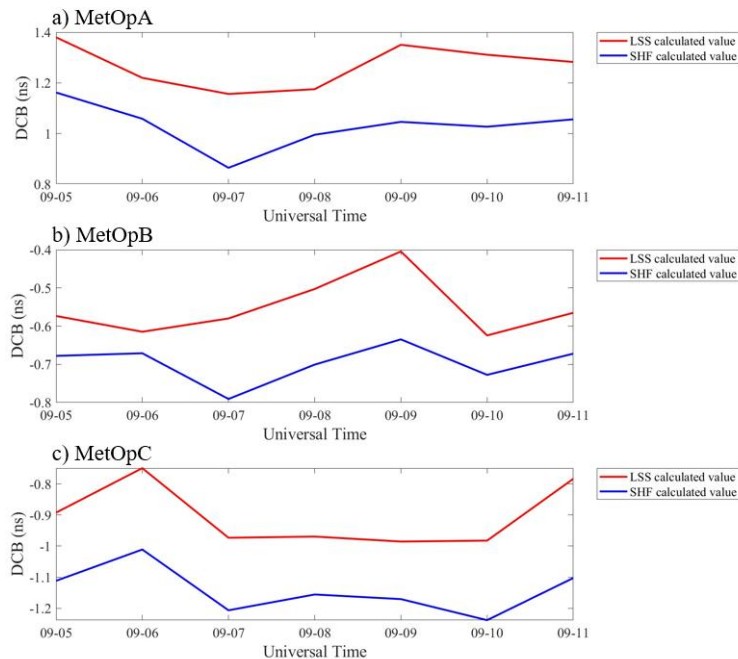

**Figure 6. DCB sequence of MetOp satellites GPS receivers from September 5 to September 11, 2021. The red line represents the receiver DCB estimated through LSS, and the blue line is the receiver DCB estimated through SHF.**





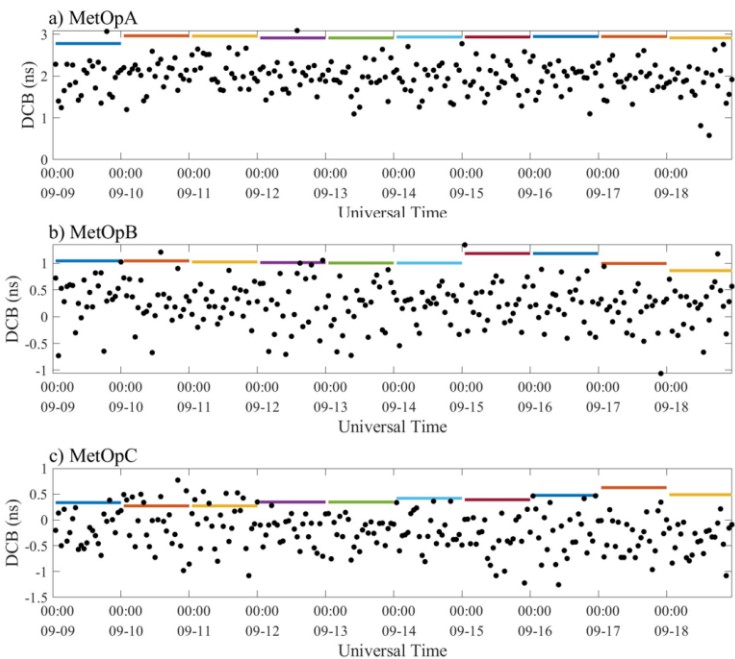

**Figure 7. Distribution of DCB time series. The lines are the reference value of MetOpA, MetOpB, and MetOpC from CDAAC, and the scatters are the calculated DCB based on LSS method. Different colors of lines represent different days reference value from CDAAC.**

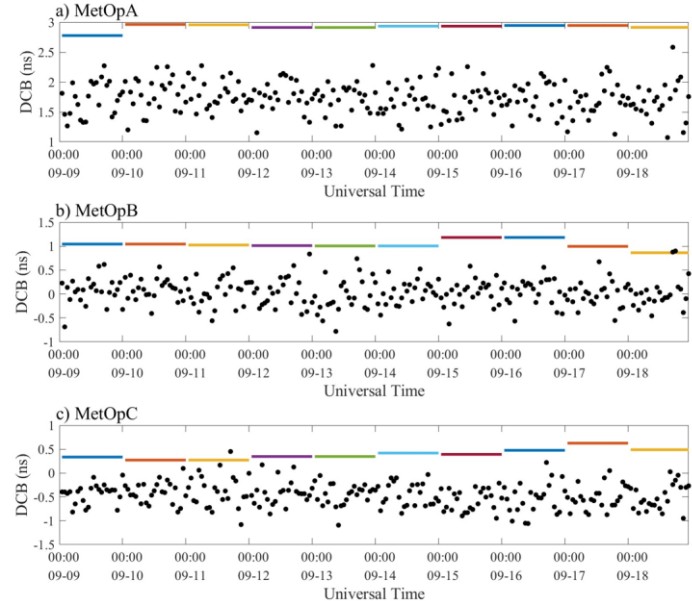

**Figure 8. Distribution of DCB time series. The lines are the reference value of MetOpA, MetOpB, and MetOpC from CDAAC, and the scatters are the calculated value by SHF method. Different color of lines represents different days reference value from CDAAC.**





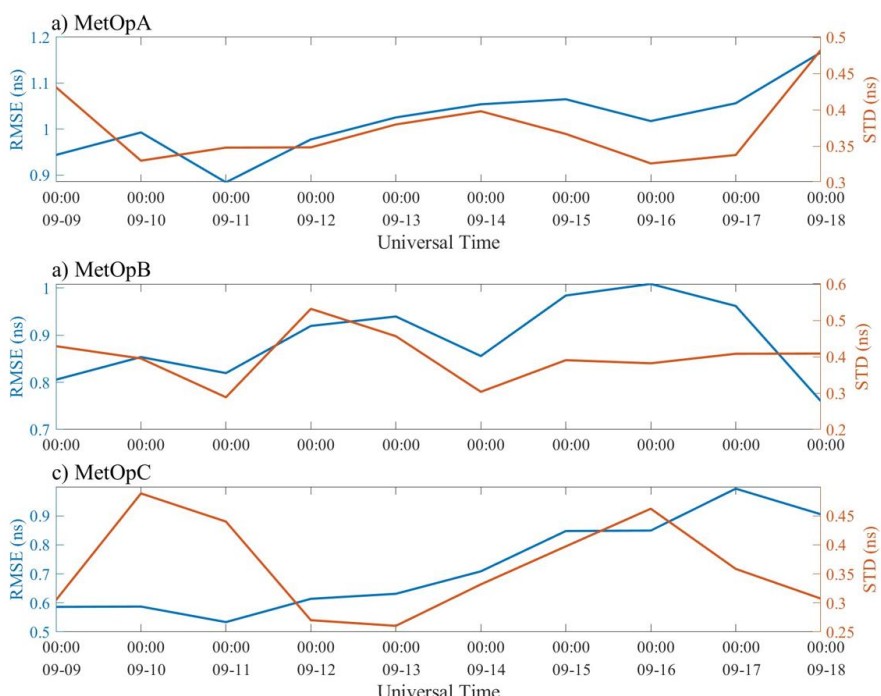

**Figure 9. RMSE and STD of DCB from LSS assumption for MetOpA, MetOpB and MetOpC, respectively.**

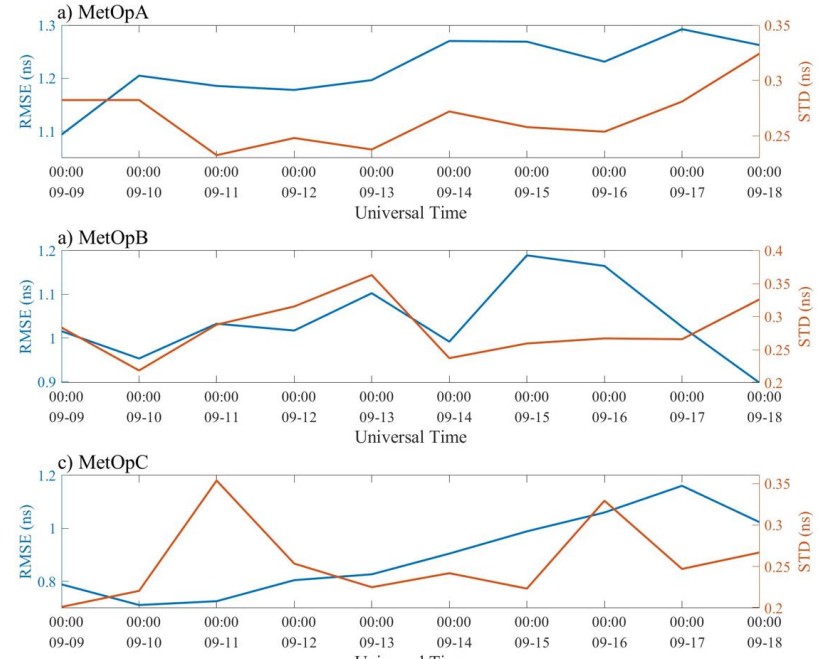

**Figure 10. RMSE and STD of DCB calculated by SHF assumption for MetOpA, MetOpB and MetOpC, respectively.**



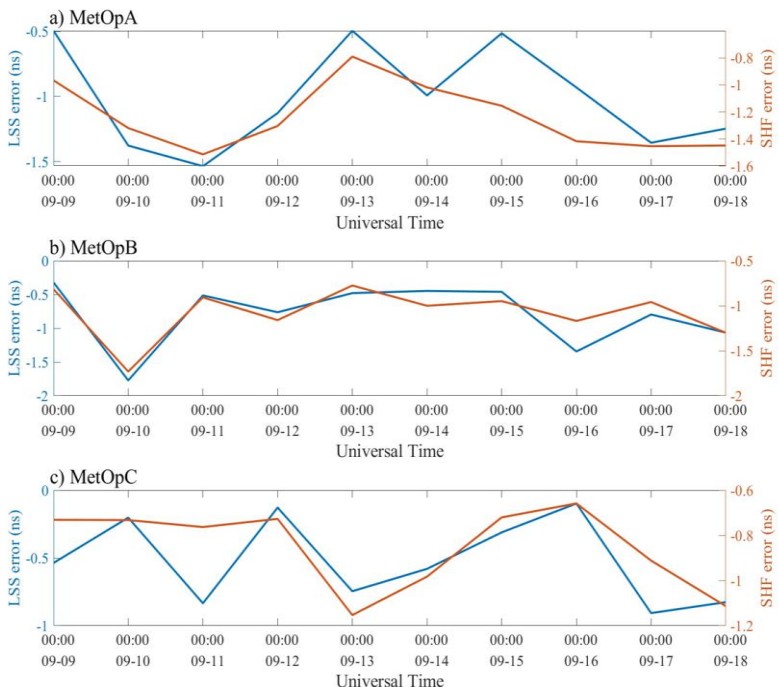

**Figure 11. Mean absolute DCB value from LSS assumption and SHF method for MetOpA, MetOpB, and MetOpC, respectively.**

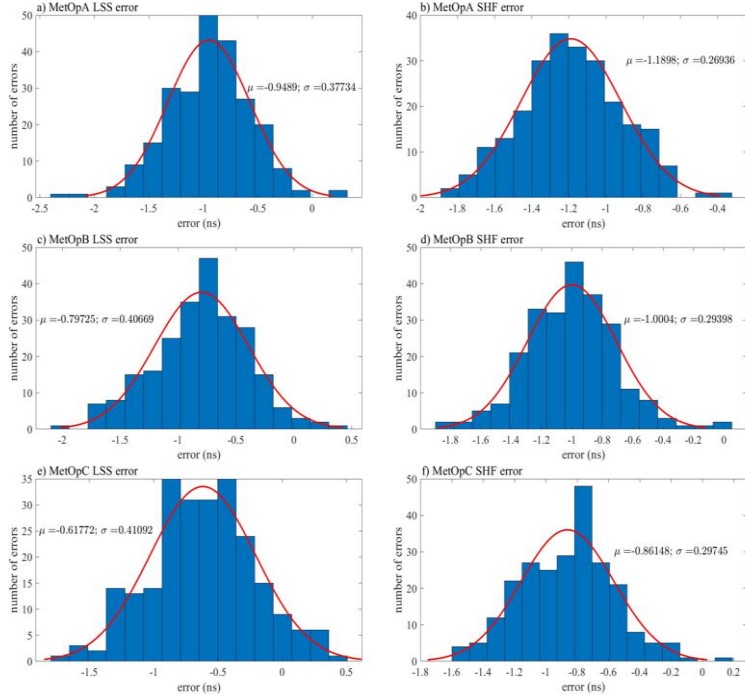

**Figure 12. Frequency statistics of LSS and SHF error numbers with LSS assumption and SHF method based on MetOpA, MetOpB, and MetOpC data during 20190909-20190918, respectively.**





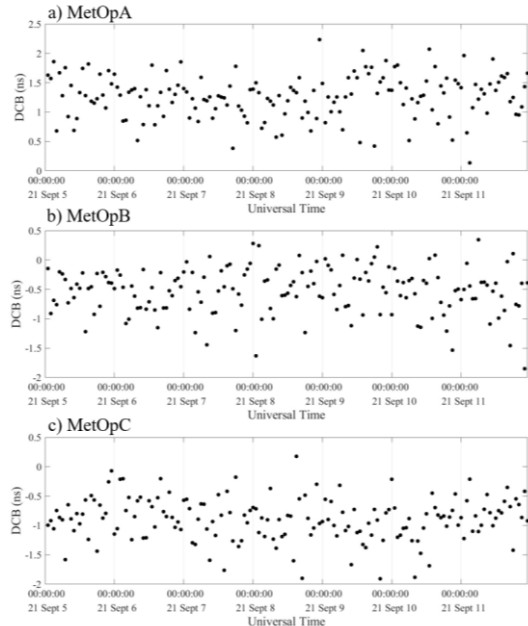

**Figure 13. Distribution of DCB time series. The scatters are the calculated values by LSS method (No reference value from CDAAC).**

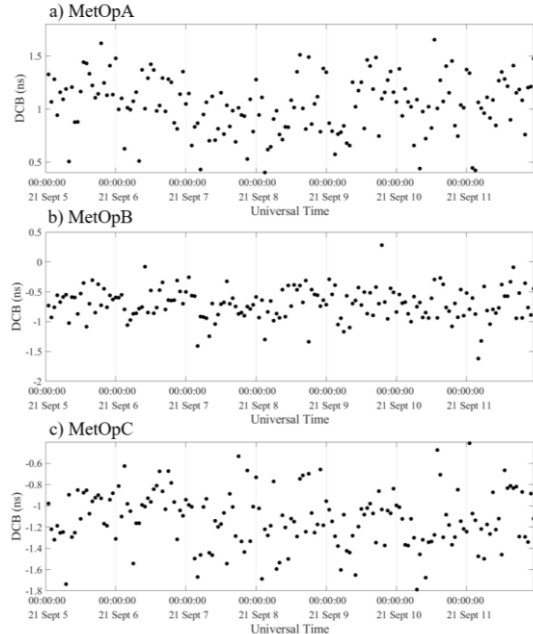

**Figure 14. Distribution of DCB time series. The scatters are the calculated values by SHF method (No reference value from CDAAC).**




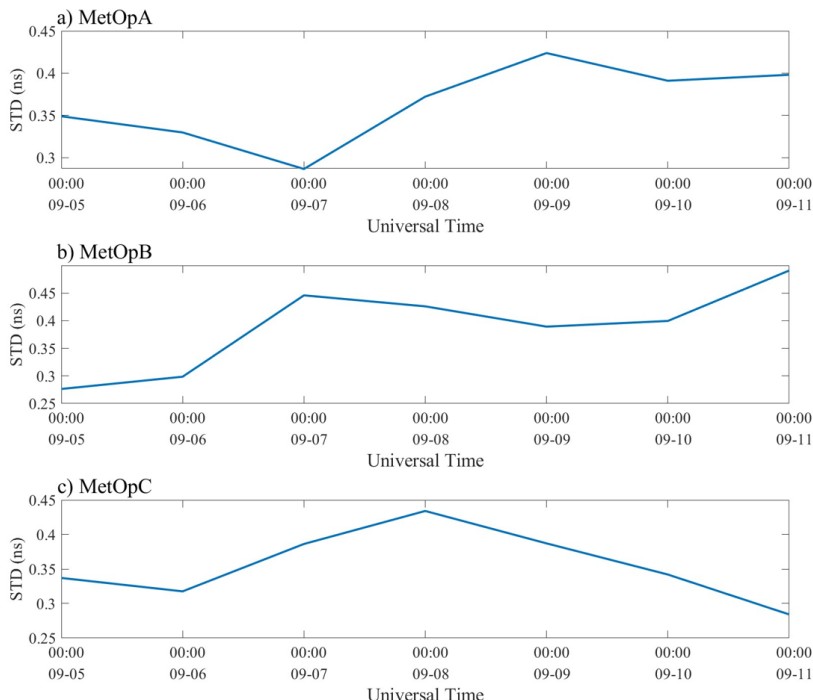

**Figure 15. STD of DCB from LSS method for MetOpA, MetOpB and MetOpC, respectively.**

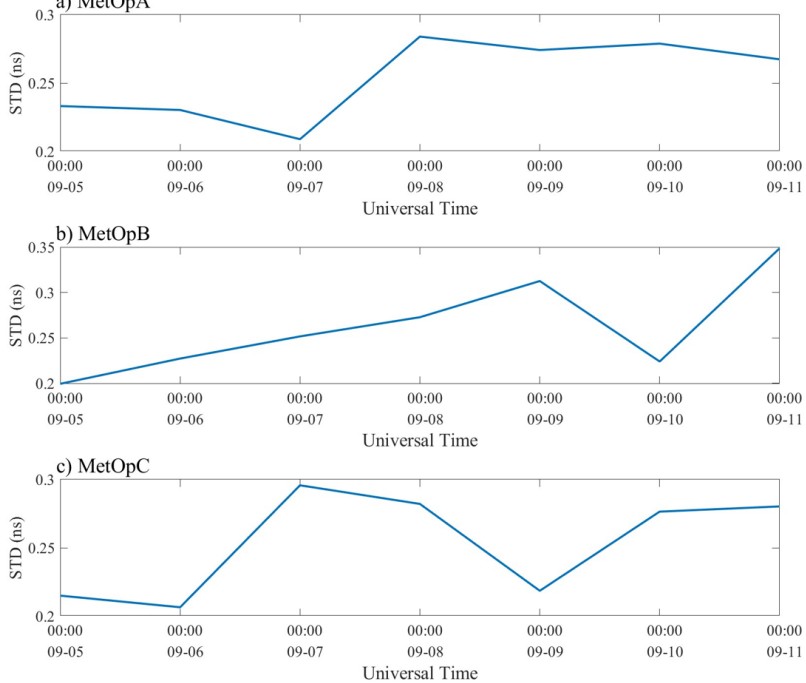

**Figure 16. STD of DCB from SHF method for MetOpA, MetOpB and MetOpC, respectively.**





**Table 1 Error analysis for different LEO satellites and different data source.**

|  |  | solar quiet days | | | solar active days | | |
|---|---|---|---|---|---|---|---|
|  |  | MetOpA | MetOpB | MetOpC | MetOpA | MetOpB | MetOpC |
|  | CDAAC | 2.92 | 1.04 | 0.40 | no available data | | |
|  | LSS | 2.78 | 1.01 | 0.57 | 1.27 | -0.55 | -0.91 |
| Mean value (ns) | SHF | 2.76 | 1.05 | 0.55 | 1.03 | -0.70 | -1.14 |
|  | CDAAC | 0.05 | 0.09 | 0.11 | no available data | | |
|  | LSS | 0.08 | 0.07 | 0.04 | 0.09 | 0.08 | 0.10 |
| STD (ns) | SHF | 0.06 | 0.07 | 0.04 | 0.09 | 0.05 | 0.08 |