# Peer review of "Estimation and evaluation of hourly MetOp satellites GPS DCBs with two different methods"

_Annales Geophysicae, 2023_

## Author Comment (AC1)

**Reviewer 1:**

Thanks for your comments and suggestions, and here is our response.

1) In the abstract, I suggest, within the sentence "Differential code bias (DCB) is one of Global Positioning System (GPS) errors, which affects the calculation of total electron content (TEC) and ionospheric modeling." to replace "which affects" by "which can affect" or "which typically affect" because there are approaches to model and compute the TEC without using pseudoranges, not requiring then to take care on DCBs. They are being used for a long time (Hernández-Pajares et al. 1999), and also recently from the GPS POD MetOp measurements only (Hernández-Pajares et al. 2023).
References:
Hernández-Pajares, M., Juan, J. M., & Sanz, J. (1999). New approaches in global ionospheric determination using ground GPS data. Journal of Atmospheric and Solar-Terrestrial Physics, 61(16), 1237-1247.
Hernández-Pajares, M., Olivares-Pulido, G., Hoque, M. M., Prol, F. S., Yuan, L., Notarpietro, R., & Graffigna, V. (2023). Topside Ionospheric Tomography Exclusively Based on LEO POD GPS Carrier Phases: Application to Autonomous LEO DCB Estimation. Remote Sensing, 15(2), 390.

Thanks for your suggestion. We have changed this in our revised MS.

2) The assessment is based on the comparison of the receiver DCBs with the reference value provided by COSMIC Data Analysis and Archive Center (CDAAC) for the MetOp A, B and C. Why are you not comparing as well the transmitter DCBs with plenty of external determinations from ground GPS data? They can be obtained from the MetOp POD GPS data only, as it has been recently done in Hernández-Pajares et al. 2023 (see reference above).

Thanks for your comments. In the reference you mentioned, the transmitter DCBs are the GPS DCB. In this study, we focus on the LEO DCBs, which are different from the GPS DCBs. Besides, to improve the accuracy of hourly DCB estimation, we used the GPS DCB products by DLR.

3) In page 2, lines 31-32, the sentence "The accuracy can be as large..." should be replaced by "The errors can be as large..." or similar.

Thanks for your suggestion. We have changed this in our MS.

4) In Equation (9) the magnitude in the DCB is changed, regarding equations (3) and (8), from range (meters) to time delay (s), by means of the explicit light speed term. A different notation between both expressions of DCBs should be used to avoid confusion for the reader between both sets of equations.
Thanks for your suggestion. We have changed this in our MS.

5) At the beginning of page 7, it is indicated that "In fact, it is also possible to estimate the GPS and receiver DCB simultaneously through certain constraints (Liu et al., 2020)." A comment on the fact that both GPS and MetOp receiver DCBs can be separately estimated from the MetOp POD GPS data only (Hernández-Pajares et al. 2023) is missed in this last paragraph of section 2.5.

Thanks for your comments. In this study, our method is to incorporate GPS DCB directly as a known quantity into the formula. For the reference, they estimated the daily GPS DCBs using tomography, which is different from our study. We have revised the words in the revised MS.

6) [Major point] The overall results and corresponding assessment is exclusively based on the receiver DCBs, compared with the reference value provided by COSMIC Data Analysis and Archive Center (CDAAC) for the MetOp A, B and C. The authors don't provide any information on the VTEC results. And this is an important lack in this work, in the experience of this reviewer, because the feasibility of the obtained VTEC (such as no negative values, very small values at local winter high latitude regions, maximum close to the equator) is a crucial information to assess the quality of the studied approach, mapping and DCBs included.

Thanks for your nice comments and good suggestion. Many previous studies were mainly focused on the DCB estimation of ground-based GNSS receivers, and VTEC maps were compared with the other existing VTEC products. However, in this study, we have focused on the space-borne LEO satellite GPS hourly DCBs estimation and comparison between different methods. For the quality of the studied approach, we have compared our estimated LEO GPS receiver DCBs results with the products from CDAAC directly. In the future, we will further estimate and study the high frequency plasmaspheric TEC (PTEC).

---

## Author Comment (AC2)

**Reviewer 2**

Thanks for your comments and suggestions, and here is our response.

This study delves into the estimation of Differential Code Bias (DCB) for the MetOP satellite. Both LSS and SHF methods were utilized to estimate the DCB for periods of one day and one hour. Despite the substantial significance of this topic for highly dynamic low Earth orbit (LEO) satellites, the manuscript presents several concerns:

1. The manuscript acknowledges the correlation between ionospheric conditions and DCB estimation; however, it lacks a comprehensive exploration of this relationship within the experimental section and estimation methodology.

5. Insufficient discussion revolves around the relationship between LEO DCB estimation, ionospheric variations, and spatial environmental changes induced by the dynamic nature of LEO satellites.

Thanks for your good comments or suggestion. Our results showed that the STD of the DCB in the active solar days is larger than this in the quiet solar days, which means that the DCBs are more stable in the quiet solar days. The details are presented in Table 1.

2. Enhancements are needed to ensure logical coherence within the introduction section for enhanced clarity and expression.

Further refinement of the manuscript is imperative, particularly in the sections concerning experimental design, results analysis, and discussion. These refinements are essential to address the a fore mentioned concerns, thereby enhancing the overall logical coherence and credibility of the article.

Thanks for your comments. We have revised the introduction in the MS to make it more logical.

3. In lines 204-205, it is stated that "DCB is assumed to remain constant within a day, with daily DCB estimation values for different time periods calculated and depicted in Figure 4 and Figure 5." Nonetheless, Figure 4 seems detached from this statement. Considering the context, Figure 4 would be more appropriately situated around line 159. This entails modifying the reference from Figure 3 in line 159 to Figure 4.

Thanks for your suggestion, and we have changed that in the MS.

4. The manuscript assesses outcomes using reference values from the COSMIC Data Analysis and Archive Center (CDAAC). Nevertheless, line 246 implies that these reference values lack precision, prompting inquiries about the practical implications of the entire analysis process.

Thanks for your comment. For the reference values from CDAAC, they are estimated by spherical symmetry assumption and there are only daily values. We imprecisely wrote these reference values with lack precision, and we have corrected our description in the revised MS.

---

## Author Response (AR2)

Reviewer 2:

The revised manuscript has made progress, but the author does not seem to have fully addressed all the issues I raised previously, such as the error in the image numbering referenced in the main text.

Thanks for your comments. We have checked and revised the Figure numbers (line 203) in the revised manuscript.